



# The dynamic of ion Bernstein-Greene-Kruskal holes in plasmas with regularized $\kappa$-distributed electrons

Qiu Ping Lu[1,2], Cai Ping Wu[1,2], Hui Chen[1,2], Xiao Chang Chen[1,2], and San Qiu Liu[1,2]

[1]Department of Physics,School of Physics and Materials Science,Nanchang University,Nanchang 330031,China
[2]JiangXi Province Key Laboratory of Fusion and Information Control,Nanchang University,Nanchang 330031,China

**Correspondence:** Hui Chen (hchen61@ncu.edu.cn)

**Abstract.** The dynamics of ion holes (IHs) in plasmas where electrons follow the regularized Kappa distribution (RKD) and ions follow the Maxwellian distribution (MD) are investigated based on the Bernstein-Greene-Kruskal (BGK) method. The results show that the depth of the IHs, the allowed combination of width and amplitude to support physically plausible IHs equilibrium depend on the spectral index $\kappa_\mathrm{e}$ and cut-off parameter $\alpha$ of the distribution function. That is, with increasing values of the spectral index $\kappa_\mathrm{e}$ and cut-off parameter $\alpha$, the IHs formed become deeper and allow a larger permissible region of width and amplitude. In contrast, with decreasing values of the spectral index $\kappa_\mathrm{e}$ and cut-off parameter $\alpha$, the IHs formed become shallower and have a smaller allowed range of width and amplitude. The present work may contribute to the comprehension of the nonlinear structures in plasmas system where non-thermal particles are found.

## 1 Introduction

Coherent bipolar electric field structures with positive and negative polarity distributions are widely presented structures that can be observed in the near-Earth plasmas (Bale et al., 1998; Matsumoto et al., 2003; Lakhina et al., 2011), space plasmas (Kakad et al., 2016; Goodrich et al., 2018), astrophysical plasmas (Vasko et al., 2018; Wang et al., 2020), laboratory plasmas (Saeki et al., 1979; Matsumoto et al., 2021) and so on. From a dynamic viewpoint, Bernstein et al. (1957) gave a steady-state solution for a coherent bipolar electric field structure with positive and negative polarity distributions in the one-dimensional stationary Vlasov-Poisson equation. The coherent bipolar electric field structures were later referred to as the Bernstein-Greene-Kruskal (BGK) mode (Temerin et al., 1982). Electrons with appropriate energy, associated with the positive monopolar wave potential, can be trapped in the potential well, forming what is known as electron holes (EHs) which have lower electron number density at the centre of phase space compared to the background plasmas (Aravindakshan et al., 2020; Hutchinson, 2017). Likewise, ions of appropriate energy and correlated with the negative monopolar wave potential can also be trapped, leading to the formation of ion holes (IHs) characterized by lower number density of ions at the centre of phase space compared to the background plasmas (Aravindakshan et al., 2018a, b, 2021, 2022).

The pseudopotential method (Gurevich, 1968; Sagdeev et al., 1969) and BGK method (Aravindakshan et al., 2018a, 2022) are usually used to study IHs. In the pseudopotential approach, the distribution functions of the trapped and passing particles are pre-assumed in order to obtain the electrostatic potential equation (Schamel, 1971; Bujarbarua et al., 1981; Schamel,





1982). Whereas in the BGK approach, one assumes the form of electrostatic potential and the distribution function of passing

particle to give the trapped ion distribution function (Aravindakshan et al., 2020, 2018a, b, 2021, 2022). Great efforts have

been devoted to study the dynamic of IHs in plasmas based on the BGK method. For example, in 2020, Aravindakshan et

al. (2020) proposed a novel BGK theory regarding IHs and applied it to the observations of the Magnetospheric Multiscale

Spacecraft (MMS), found that the theoretical predictions matched the observed results. In 2021, they investigated the influence

of different plasma parameters on the formation and structure of IHs (Aravindakshan et al., 2021). In 2022, Aravindakshan et

al. (2022) explored the impact of the ion-to-electron temperature ratio on the IHs in dusty plasmas. In 2023, they studied the

effect of the ion-to-electron temperature ratio, Mach number, and the shape of the electron distribution on the properties of IHs

(Aravindakshan et al., 2023). Most of the studies mentioned above have postulated that the electrons follow the Maxwellian

distribution (MD) or standard Kappa distribution (SKD).

MD can describe the electron distribution in thermal equilibrium very well. However, it is not suitable for the description of

the electron distribution in non-thermal equilibrium (Aravindakshan et al., 2018a; Li et al., 2023). It is well known that SKD is

appropriate for the situation where the spectral index $\kappa_e$ exceeds 1.5, but not for the case of $\kappa_e$ between 0 and 1.5. This is due

to the fact that the effective temperature should be finite and positive. In addition, it manifests diverging velocity moments and

a non-extensive entropy (Lazar et al., 2021). Besides these major limitations of SKDs, there is another observation limitation,

namely the inability of SKD to explain velocity distributions that are harder than $v^{-5}$ (Haas et al., 2023). Gloeckler et al. (2012)

analysed the superthermal tail using hourly-averaged proton velocity distributions at 1 AU and observed the distributions with

harder tails. These measurements also show that the kappa can be close to 2 or less than 2. For example, distributions of solar

wind electrons and solar energetic particles distributions have been found to have kappa values as low as 2 (Pierrard et al., 2022;

Oka et al., 2013). In order to overcome the limitations of SKDs, in 2018, Scherer et al. (2018) introduced exponential cutoff

in SKDs and named SKDs with exponential cutoff as regularized Kappa distribution (RKD). For RKD, the velocity moments

of any order are convergent when $\kappa > 0$. More recently, the characteristics of nonlinear structures such as electrostatic solitary

waves, ion temperature gradient (ITG) modes, and EHs in plasmas where electrons follow the RKD have been investigated.

For instance, Liu (2020) conducted research on the existence conditions and properties of ion acoustic solitons (IAS). It was

shown that $\alpha$ and $\kappa_e$ are key factors influencing the existence and the width of IAS. Lu et al. (2021) found that when the cut-off

index $\alpha$ increases, the amplitude and width of the solitary wave decrease and its propagation velocity is smaller than that of

the electron containing the SKD. Liu et al. (2021) observed that the linear behavior of Langmuir wave (LW) is greatly changed

by the parameters $\alpha$ and $\kappa_e$ and when $\kappa_e < 1.5$, the damping rate of LW is much larger than that with Maxwellian distributed

electrons. Zhou et al. (2022) examined linear and nonlinear ITG modes using a fluid model and discovered that the ITG modes

become unstable as the parameters $\alpha$ and $\kappa_e$ increase. Haas et al. (2023) derived the pseudopotential in the weakly nonlinear

limit based on the pseudopotential method, found the most prominent solution to the resulting Poisson equation, discussed

the drifting, non-drifting, oscillating and non-oscillating solutions, analysed the linear dispersion relations, and discovered

the existence of a high-frequency Langmuir mode, a low-frequency electron acoustic mode and the structure is recovered

as a standard electron hole structure when $\kappa \gg 1$. To the best of our knowledge, the IHs in unmagnetized and collisionless

electron-ion plasmas utilizing the BGK method with RKD electrons have not yet been explored.





In the present work, the dynamic of IHs in plasmas are investigated in the context of RKD electrons based on the BGK approach. The paper is organized in the following way. In Section 2, the fundamental equations for ion BGK holes with RKD electrons and the relation between width and amplitude are obtained. The analysis and discussion of the numerical results are presented in Section 3. Finally, the summary and main conclusions of the work are given in Section 4.

## 2 Theoretical model

Let us consider a one-dimensional unmagnetized and collisionless plasma system consisting of ions and electrons which obey the RKD. The Vlasov and Poisson equations for the ions are given by

$$\left(\frac{\partial}{\partial t} + V_{\mathrm{i}}\frac{\partial}{\partial x} - \frac{q_{\mathrm{i}}}{m_{\mathrm{i}}}\frac{\partial \Phi}{\partial x}\frac{\partial}{\partial V_{\mathrm{i}}}\right) f_{\mathrm{i}} = 0, \tag{1}$$

$$\frac{\mathrm{d}^2 \Phi}{\mathrm{d}x^2} = \frac{-(q_{\mathrm{e}}N_{\mathrm{e}} + q_{\mathrm{i}}N_{\mathrm{i}})}{\varepsilon_0}, \tag{2}$$

where $f_{\mathrm{i}}$, $q_{\mathrm{i}}$, $V_{\mathrm{i}}$, $m_{\mathrm{i}}$ are the distribution function, charge, velocity and mass of ions, respectively. $x$ is in the unit of the ion Debye length $\lambda_{\mathrm{di}} = \left(\frac{k_{\mathrm{B}}T_{\mathrm{i}}}{\epsilon_0 N_0 e^2}\right)^{\frac{1}{2}}$. $\Phi$ represents the electrostatic potential, and $\varepsilon_0$ denotes the vacuum permittivity, $N_{\mathrm{e}}$ and $N_{\mathrm{i}}$ represent the number densities of electrons and ions, which are assumed to be equal at a value of $N_0$. The charge of electrons is $q_{\mathrm{e}} = -e$, while the charge of ions is $q_{\mathrm{i}} = e$. Working in a coordinate system where the IHs in a stationary state is considered, so that the quantities are independent of time. In this case, Eqs.(1) and (2) can be simplified to the following dimensionless

forms (Bernstein et al., 1957):

$$\left(v\frac{\partial}{\partial x} + \frac{1}{2}\frac{\partial \phi}{\partial x}\frac{\partial}{\partial v}\right) f_{\mathrm{i}}(x,v) = 0, \tag{3}$$

$$\frac{\mathrm{d}^2 \phi}{\mathrm{d}x^2} = n_{\mathrm{e}} - n_{\mathrm{i}}, \tag{4}$$

where $\phi$, $v$, $n_{\mathrm{i}}$ are the electrostatic potential, ion velocity, ion number density normalized as $\frac{k_{\mathrm{B}}T_{\mathrm{i}}}{e}$, $v_{\mathrm{th,i}} = \left(\frac{2k_{\mathrm{B}}T_{\mathrm{i}}}{m_{\mathrm{i}}}\right)^{\frac{1}{2}}$, $N_0$, with

$k_{\mathrm{B}}$ and $T_{\mathrm{i}}$ being the Boltzmann constant and ion temperature, respectively.

In the electrostatic potential, the regularized $\kappa$-distributed electrons can be expressed as (Liu, 2020):

$$n_{\mathrm{e}} = \exp\left(\alpha^2 \phi\right)\left(1 - \frac{\phi}{\kappa_{\mathrm{e}}}\right)^{-\kappa_{\mathrm{e}}+1/2}\frac{U\left[\frac{3}{2}, \frac{3}{2} - \kappa_{\mathrm{e}}; \alpha^2\kappa_{\mathrm{e}}\left(1 - \frac{\phi}{\kappa_{\mathrm{e}}}\right)\right]}{U\left(\frac{3}{2}, \frac{3}{2} - \kappa_{\mathrm{e}}; \alpha^2\kappa_{\mathrm{e}}\right)}, \tag{5}$$

where $\kappa_{\mathrm{e}}$, $\alpha$, and $U(a,c;x)$ are, respectively, the electron spectral index, the exponential cutoff parameter and the Kummer function. In fact, in the case of $\alpha \to 0$ and $\kappa_{\mathrm{e}} > 1.5$, Eq.(5) is transformed into the SKD. When $\alpha \to 0$ and $\kappa_{\mathrm{e}} \to \infty$, Eq.(5)

reduces to the MD. Nevertheless, the distribution of electrons with the SKD in the potential field will be valid within the region




where $\kappa_e$ is greater than 0 and $\alpha \neq 0$. In this paper, we extended the SKD to the range of $\kappa > 0$ to demonstrate the impact of the cut-off parameter $\alpha$ on the properties of ion BGK holes.

Under the small amplitude approximation, Eq.(5) can be approximated by Taylor expanding for $\phi/\kappa_e \ll 1$ (Liu, 2020):

$$n_e \approx 1 + A\frac{\phi}{\kappa_e} + \frac{1}{2}B\frac{\phi^2}{\kappa_e{}^2} + \cdots,$$ (6)

where

$$A = \kappa_e - \frac{1}{2} + \kappa_e \alpha^2 \left[1 + \frac{3}{2}\frac{U\left(\frac{5}{2}, \frac{5}{2} - \kappa_e; \alpha^2\kappa_e\right)}{U\left(\frac{3}{2}, \frac{3}{2} - \kappa_e; \alpha^2\kappa_e\right)}\right]$$

and

$$B = \kappa_e{}^2 - \frac{1}{4} + 2\alpha^2\left(\kappa_e - \frac{1}{2}\right)\kappa_e\left[1 + \frac{3}{2}\frac{U\left(\frac{5}{2}, \frac{5}{2} - \kappa_e; \alpha^2\kappa_e\right)}{U\left(\frac{3}{2}, \frac{3}{2} - \kappa_e; \alpha^2\kappa_e\right)}\right] +$$
$$\kappa_e{}^2\alpha^4\left[1 + 3\frac{U\left(\frac{5}{2}, \frac{5}{2} - \kappa_e; \alpha^2\kappa_e\right)}{U\left(\frac{3}{2}, \frac{3}{2} - \kappa_e; \alpha^2\kappa_e\right)} + \frac{15}{4}\frac{U\left(\frac{7}{2}, \frac{7}{2} - \kappa_e; \alpha^2\kappa_e\right)}{U\left(\frac{3}{2}, \frac{3}{2} - \kappa_e; \alpha^2\kappa_e\right)}\right].$$

It is noteworthy that under the condition of $\phi/\kappa_e \ll 1$, the value of $n_e$ in Eq.(5) will not diverge as $\kappa_e$ approaches zero.

The normalized form of MD for the ions takes the form:

$$f_i(v) = \frac{1}{\sqrt{\pi}}\exp\left(-v^2\right).$$ (7)

Introducing the normalized total particle energy,

$$w = \frac{1}{2}\left(v^2 + \phi\right),$$ (8)

then Eq.(7) can be transformed into the following form:

$$f_i(w) = \frac{1}{\sqrt{\pi}}\exp\left(-2w + \phi\right),$$ (9)

where $f(x, v)\,dv = f(w)\,dw/\sqrt{2w - \phi}$ has been used.

When ions run into a negative electrostatic potential, some will be trapped while others pass through. Therefore, there are two types of ions: the passing ions $n_p$ and trapped ions $n_{tr}$. Hence, the normalized Poisson equation (4) can be written in the following form:

$$\frac{d^2\phi}{dx^2} = n_e - n_p - n_{tr}.$$ (10)

When $w > 0$, ions are trapped within the range of $\left[-\sqrt{-\phi}, +\sqrt{-\phi}\right]$, otherwise pass through, i.e., the integral forms for the passing ions $n_p$ and trapped ions $n_{tr}$ can be written as follows:

$$n_p = \int\limits_{-\infty}^{-\sqrt{-\phi}} f_p(x, v)\,dv + \int\limits_{+\sqrt{-\phi}}^{+\infty} f_p(x, v)\,dv$$ (11)




and

$$n_{\mathrm{tr}} = \int\limits_{-\sqrt{-\phi}}^{+\sqrt{-\phi}} f_{\mathrm{tr}}(x,v)\,\mathrm{d}v, \tag{12}$$

where $f_{\mathrm{p}}(x,v)$ and $f_{\mathrm{tr}}(x,v)$ are the distribution functions of the passing and trapped ions, respectively. Substituting Eq.(7) into Eq.(11) and after integration, the density $n_{\mathrm{p}}$ of the passing ion is

$$n_{\mathrm{p}} = 1 - \mathrm{erf}\left(\sqrt{-\phi}\right), \tag{13}$$

here, $\mathrm{erf}\left(\sqrt{-\phi}\right)$ represents the error function.

It has been revealed by spacecraft observations that Gaussian-shaped wave potential structures are commonly observed in the Earth's magnetosphere (Matsumoto et al., 1994), space and astrophysical plasmas (Williams et al., 2006). Let us hypothesize that the potential structure is of Gaussian form, i.e.,

$$\phi(x) = -\psi \exp\left(-\frac{x^2}{2\delta^2}\right), \tag{14}$$

where $\psi$ and $\delta$ denote the amplitude and width of the wave potential. By differentiating Eq.(14) twice with respect to $x$, one obtains

$$\frac{\mathrm{d}^2\phi}{\mathrm{d}x^2} = \frac{x^2\phi}{\delta^4} - \frac{\phi}{\delta^2} = -\frac{2\phi\ln\left(-\frac{\phi}{\psi}\right)}{\delta^2} - \frac{\phi}{\delta^2}. \tag{15}$$

By substituting Eq.(15) into Eq.(10), the density of trapped ions is obtained as

$$n_{\mathrm{tr}} = A\frac{\phi}{\kappa_{\mathrm{e}}} + \frac{1}{2}B\frac{\phi^2}{\kappa_{\mathrm{e}}^2} + \mathrm{erf}\left(\sqrt{-\phi}\right) + \frac{2\phi\ln\left(-\frac{\phi}{\psi}\right)}{\delta^2} + \frac{\phi}{\delta^2}. \tag{16}$$

Then substituting Eq.(16) into Eq.(12), the trapped ion distribution function $f_{\mathrm{tr}}$ can be expressed in the following form:

$$f_{\mathrm{tr}}(w) = -\frac{2A}{\pi\kappa_{\mathrm{e}}}\sqrt{-2w} - \frac{8Bw}{3\pi\kappa_{\mathrm{e}}^2}\sqrt{-2w} +$$
$$\frac{2\sqrt{2}\sqrt{-w}}{\pi\delta^2}\left[1 - 2\ln\left(-8w/\psi\right)\right] + \frac{\mathrm{e}^w I_0(w)}{\sqrt{\pi}}, \tag{17}$$

where $I_0(x)$ denotes the first class of modified Bessel functions.

In order to have physical feasibility, the resulted trapped ion distribution function must be greater than zero, then the width-amplitude relationship for stable equilibrium of IHs can be obtained as follows:

$$\delta^2 \geq \frac{6\sqrt{2}\sqrt{-w}\kappa_{\mathrm{e}}^2\left[1 - 2\ln\left(-8w/\psi\right)\right]}{6\sqrt{2}\sqrt{-w}A\kappa_{\mathrm{e}} + 8\sqrt{2}\sqrt{-w}Bw - 3\kappa_{\mathrm{e}}^2\mathrm{e}^w I_0(w)}, \tag{18}$$

which restricts both the width and amplitude of the wave potential responsible for sustaining IHs. When $\alpha = 0$ and $\kappa_{\mathrm{e}} \to \infty$, Eqs. (17) and (18) return to the results obtained by Aravindakshan et al. (2022) in the Appendix.


# 3  Analysis and discussion

In this section, the plasma parameters related to ion BGK holes are examined by analyzing Eqs.(19) and (20). Studying the trapped ion distribution function that emerges in the phase space is imperative to obtain a more comprehensive understanding

of the qualitative influence of relevant plasma parameters. In the phase space, the behavior of the trapped ion distribution functions versus $x$ and $v$ for various electron spectral indices $\kappa_e$ and cut-off index $\alpha$ are illustrated in Figs. 1, 2 and 3. It can be observed from Figs. 1 and 2 that the depth of the IHs becomes greater in response to an increased $\kappa_e$. It is also worth noting that the distribution function for both $\kappa_e = 3$ and $\kappa_e = 100$ is not significantly different when $\alpha = 0$ and $\alpha = 0.01$, it is due to the fact that when $\alpha$ tends to 0 and $\kappa_e$ is greater than 1.5, RKD tends to SKD. As $\alpha$ approaches 0 and $\kappa_e$ approaches infinity,

RKD approaches MD. From a physical standpoint, the concentration of superthermal electrons is determined by $\kappa_e$. The lower the electron spectral index $\kappa_e$, the higher the concentration of superthermal electrons, then the higher the amount of energy it carries. As a result, more energy is transferred to the ions, then the ions have enough energy to escape the potential well and not be trapped, resulting in the inhibition of the formation of ion holes, so the formed holes are shallower. In addition, in Fig. 3, it can also be clearly seen that when $\alpha$ increases, the IHs becomes deeper. This can be attributed to the fact that a large the cut-off

index $\alpha$ will lead to a decrease in the concentration of superthermal electrons. Consequently, fewer superthermal electrons will not be able to cause more ions to escape from the potential well and become passing ions. As a consequence, a greater number of ions are trapped, resulting in the formation of deeper holes.

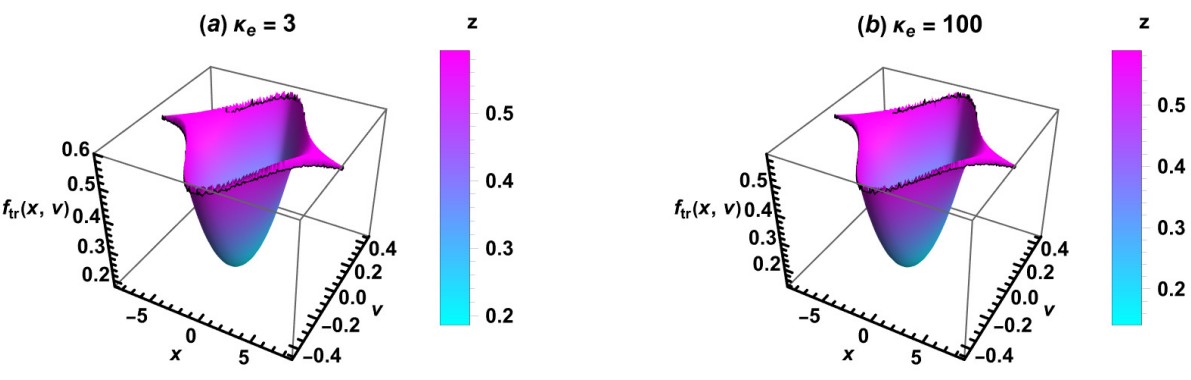

**Figure 1.** The phase space structure of the trapped ion distribution function $f_{\mathrm{tr}}(x,v)$ in $x-v$ space for different $\kappa_e$ at $\alpha = 0$, $\delta = 0.2$ and $\psi = 2$. (a)$\kappa_e = 3$, (b)$\kappa_e = 100$.

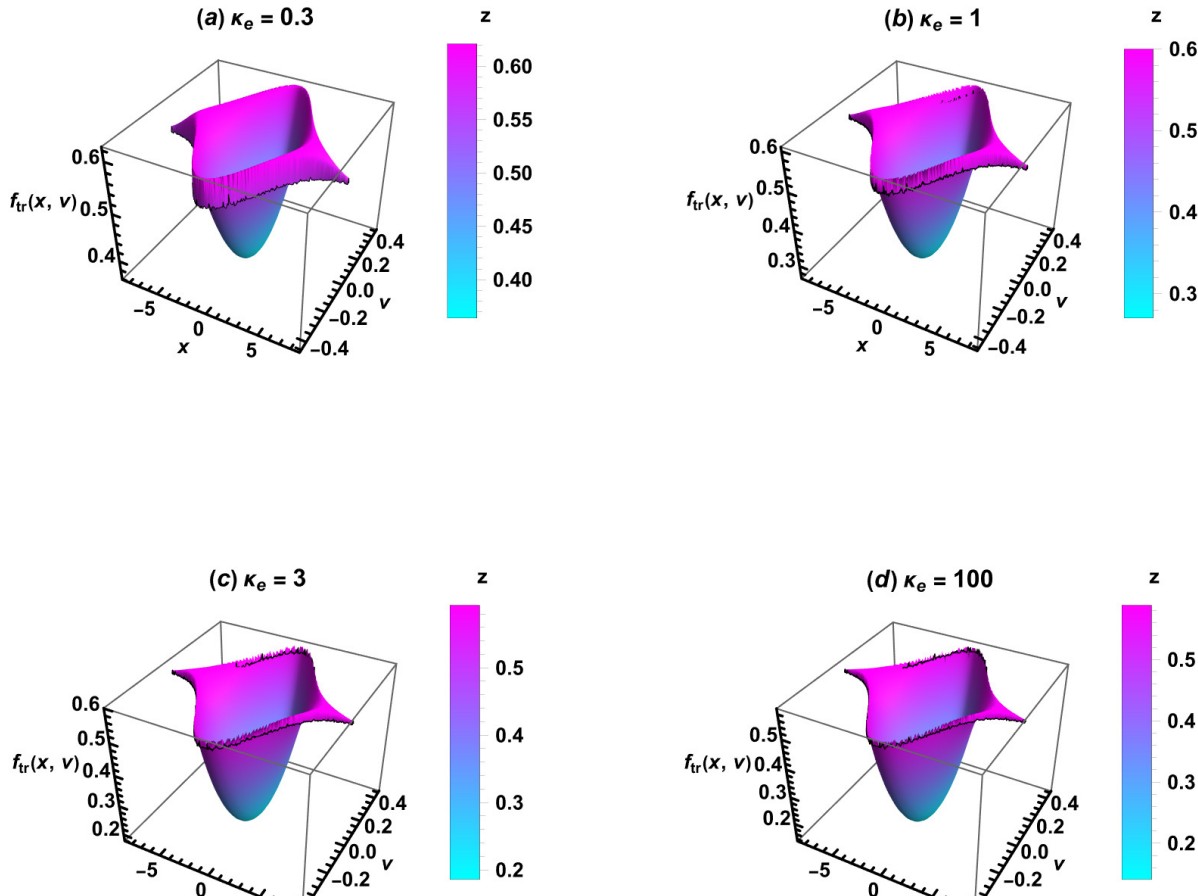

**Figure 2.** The phase space structure of the trapped ion distribution function $f_{\mathrm{tr}}(x,v)$ in $x-v$ space for different $\kappa_{\mathrm{e}}$ at $\alpha = 0.01$, $\delta = 0.2$ and $\psi = 2$. (a)$\kappa_{\mathrm{e}} = 0.3$, (b)$\kappa_{\mathrm{e}} = 1$, (c)$\kappa_{\mathrm{e}} = 3$, (d)$\kappa_{\mathrm{e}} = 100$.

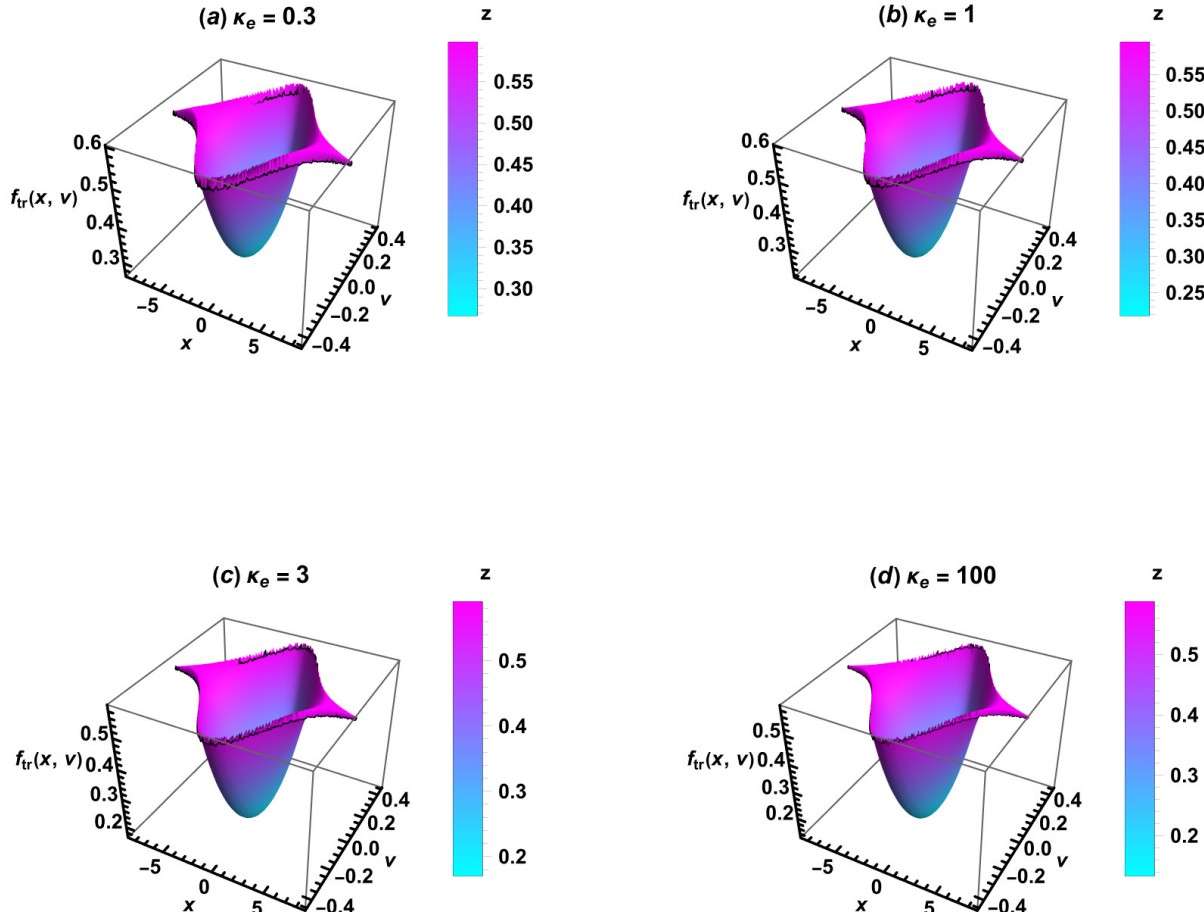

**Figure 3.** The phase space structure of the trapped ion distribution function $f_{\mathrm{tr}}(x,v)$ in $x-v$ space for different $\kappa_{\mathrm{e}}$ at $\alpha = 0.2$, $\delta = 0.2$ and $\psi = 2$. (a)$\kappa_{\mathrm{e}} = 0.3$, (b)$\kappa_{\mathrm{e}} = 1$, (c)$\kappa_{\mathrm{e}} = 3$, (d)$\kappa_{\mathrm{e}} = 100$.

For a better understanding, the variations of $f_{\mathrm{tr}}$ in relation to the energy parameter $w$ for various $\kappa_{\mathrm{e}}$ and $\alpha$ is depicted in Fig. 4. As shown in Fig. 4(a), when $w$ approaches 0, the six curves converge to the point of $f_{\mathrm{tr}}(w) = 0.58$. When $\alpha = 0$, the distribution functions for $\kappa_{\mathrm{e}} = 3$ and $\kappa_{\mathrm{e}} = 100$ almost coincide with the distribution function when $\alpha = 0.01$. In Fig. 4(a), the difference between $f_{\mathrm{tr}}$ corresponding to $w = -0.1$ and the maximum value of $f_{\mathrm{tr}}$ represents the depth of the IHs formed, from which it can be found that a higher $\kappa_{\mathrm{e}}$ leads to deeper IHs. Physically speaking, a lower $\kappa_{\mathrm{e}}$ results in an increase in the electron concentration in the high-energy tail of the SKD, which implies that more superthermal electrons are present. The presence of


more superthermal electrons means that superthermal electrons have higher energies, which allows them to escape from the
potential well and form IHs. As a result, this will lead to the formation of a shallower IH encompassing comparatively less
trapped ions. From Fig. 4(b), it can be seen that the larger $\alpha$ is, the deeper the IHs are formed. Physically, the larger $\alpha$ restricts
the presence of high-energy electrons, thereby limiting the population of superthermal electrons and enhancing the number of
trapped ions in plasmas. Consequently, a higher $\alpha$ leads to deeper IHs.

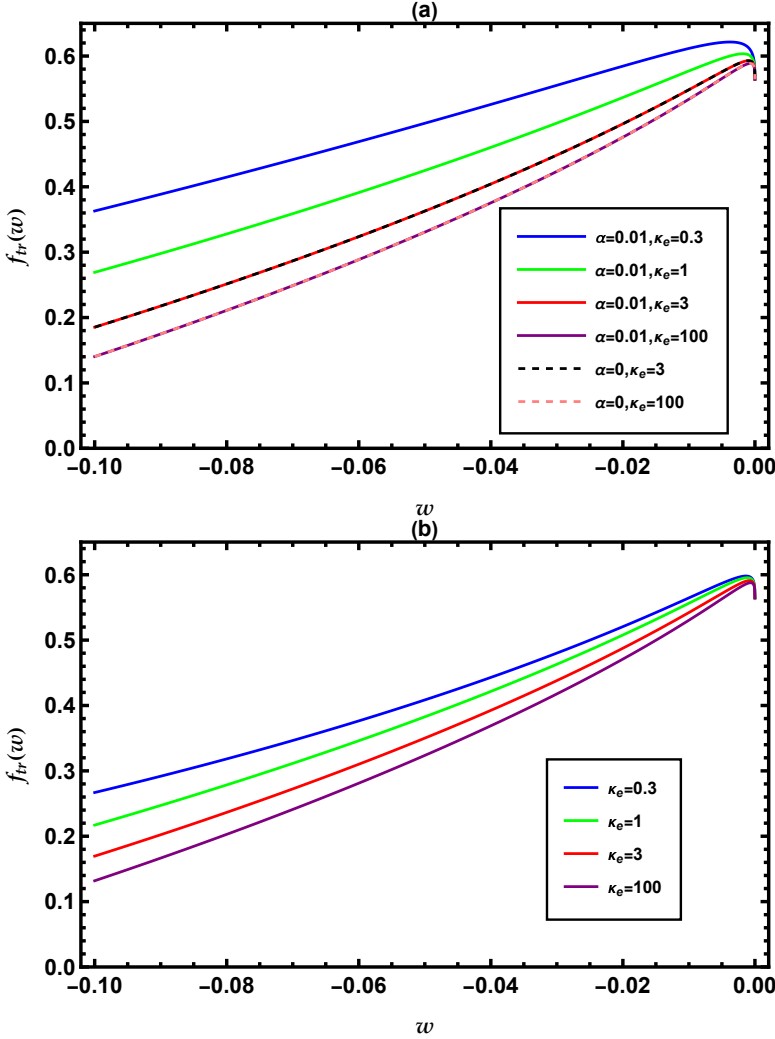

**Figure 4.** The trapped ion distribution function vs energy $w$ for different $\kappa_{\mathrm{e}}$ and $\alpha$, while keeping $\delta = 0.2$ and $\psi = 2$ as fixed. (a)$\alpha = 0$ and
$\alpha = 0.01$, (b)$\alpha = 0.1$, where the black dashed, pink dashed, blue, green, red and purple line correspond to $\kappa_{\mathrm{e}} = 3$, 100, 0.3, 1, 3 and 100,
respectively.

The impact of $\alpha$ and $\kappa_{\mathrm{e}}$ on the physically plausible region for the existence of IHs is shown in Fig. 5. It can be seen from
Fig. 5(a) that for $\alpha = 0$, the curves of $\kappa_{\mathrm{e}} = 3$ and $\kappa_{\mathrm{e}} = 100$ are almost identical with $\alpha = 0.01$. In the case of a smaller $\kappa_{\mathrm{e}}$, it


can be observed that the plausible range increases. From a physical perspective, when the $\kappa_e$ is smaller, there will be a greater number of superthermal electrons carrying more energy, resulting in a higher energy transfer to the ions. Due to the presence of a large number of energetic ions, the plausible region of the IHs becomes larger. In Fig. 5(b), one can find that when $\alpha$ is larger, the observed plausible region is smaller. Physically, as $\alpha$ decreases, there are more superthermal electrons which carry more

energy. They transfer more energy to the ions. In other words, there are a large number of high-energy ions, so the plausible region of the IHs is larger.

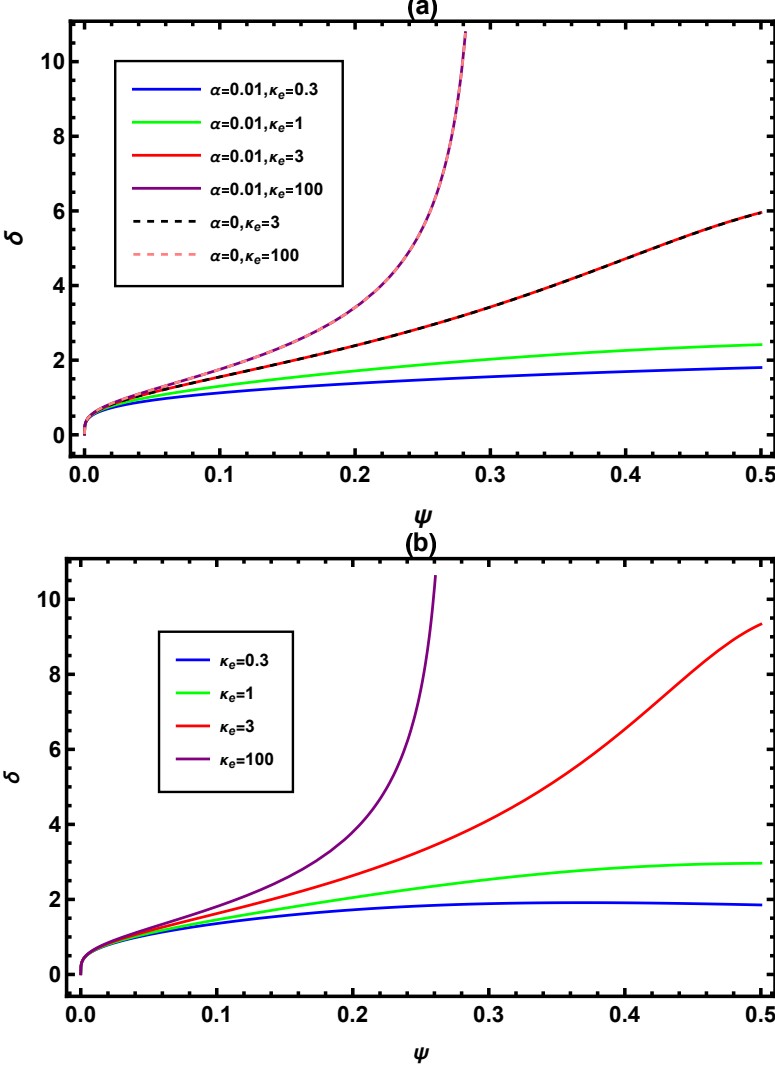

**Figure 5.** The variation of width $\delta$ vs amplitude $\psi$ for different $\kappa_e$ and $\alpha$. (a)$\alpha = 0$ and $\alpha = 0.01$, (b)$\alpha = 0.1$, where the black dashed, pink dashed, blue, green, red and purple line correspond to $\kappa_e = 3, 100, 0.3, 1, 3$ and $100$, respectively.





## 4 Conclusions

In this paper, assuming that the electrons follow RKD and ions obey MD, the kinetic properties of IHs in plasmas are investigated by using the BGK approach. The effects of the spectral index $\kappa_{\mathrm{e}}$ and cut-off parameter $\alpha$ on the structure of the

IHs are analysed. It is found that as the values of $\kappa_{\mathrm{e}}$ and $\alpha$ increase, the depth of the IHs formation increases while the region in which the IHs exist becomes smaller. Physically, when $\kappa_{\mathrm{e}}$ and $\alpha$ increase, the number of superthermal electrons decreases, more ions are trapped in the IHs, and the deeper IHs are formed. It may be stressed here that the results of present work may provide some theoretical references for the understanding of the nonlinear structures in plasmas system where non-thermal particles are found.

*Data availability.* Any data that support the findings of this study are included within the article.

*Author contributions.* Qiuping Lu conceived the work, performed the numerical calculations, and wrote the manuscript. Hui Chen contributed to the analyses. Caiping Wu, Hui Chen, Xiaochang Chen and Sanqiu Liu revised the manuscript.

*Competing interests.* The authors declare that they have no competing interest.

*Acknowledgements.* The work is supported by the Project of Scientific and Technological Innovation Base of Jiangxi Province (Grant

No.20203CCD46008), the Key R&D Plan of Jiangxi Province (Grant No.20223BBH80006) and the Jiangxi Province Key Laboratory of Fusion and Information Control (Grant No.20171BCD40005).





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
