# Peer review of "The dynamic of ion Bernstein-Greene-Kruskal holes in plasmas with regularized $\kappa$ -distributed electrons"

_Nonlinear Processes in Geophysics, 2023_

## Author Comment (AC1)

**Letter to the editor**

Dear editor,

Thank you very much for the efficient work on the manuscript entitled "The dynamic of ion Bernstein–Greene–Kruskal holes in plasmas with regularized $\kappa$-distributed electrons" (npg-2023-25). We have revised the manuscript according to the Referees' comments. The detailed description of the revisions made in the paper is included in the response. For more clearly reading, the highlighted manuscript where all the revises are marked with yellow is also delivered. We appreciate for your warm work earnestly, and hope our revisions are acceptable. If you have any queries, please don't hesitate to contact us.

Thank you again.
Yours sincerely,
Qiuping Lu, Caiping Wu, Hui Chen, Xiaochang Chen and Sanqiu Liu
* * *
**Reply to Referees**

Dear Referee,

Deeply thanks to you for your constructive and very helpful comments. Our manuscript has been revised according to your comments, and the point-by-point replies for your comments are as follows. Detailed revises are marked with yellow in the revised version of manuscript. We appreciate for your warm work earnestly, and hope that the correction will meet with approval.

Thank you again for your comments and suggestions

Sincerely,
Qiuping Lu, Caiping Wu, Hui Chen, Xiaochang Chen and Sanqiu Liu.

**Point-by-point replies to the referee 1's comments as follows,**

**Question 1:** Observational Evidence of RKD: It is crucial for the manuscript to include specific observational results that substantiate the physical existence of conditions where kappa less than or equal to 1.5. While the mathematical premise stands, empirical evidence would solidify the paper's relevance. For instance, statistical surveys of the solar wind and magnetosphere have revealed kappa values between 2-6, but intervals with k < 2 are not uncommon during dynamic processes like magnetic reconnection.

Referencing such observations is essential to substantiate the regimes where their model is applicable. The authors should also discuss whether there are specific instances in space plasma environments where this condition is observed? Providing such observational instances would significantly elevate the paper's practical implications in the field.

**Reply:** We appreciate for your constructive suggestion and quite agree that it is crucial for the manuscript to include specific observational results that substantiate the physical existence of conditions where kappa less than or equal to 1.5 and that the provision of examples of such observations will significantly elevate the practical implications of the paper in the field. I am sorry that, at the moment, we have not found a concrete example of an observation and specific instances in space plasma environments where this condition is observed, but I hope that this study will provide a theoretical reference for the results of observing such a situation in space plasma. What's more, we have added some references of observation on page 2 line 20-22, which are about the particle distribution function close to and below 2.

**i.e.:** There are evidences that kappa values may be as low as 1.63 and 2 for particle distributions in the outer heliosphere (Heerikhuisen et al., 2008; Zirnstein et al., 2017). It has also been observed that the kappa values of the particle distributions may be lower than 1.5 (Desai et al., 2020, 2016) or even as low as 0.2 (Vasko et al., 2017).

**Question 2:** While the focus on unmagnetized and collisionless electron-ion plasmas using the BGK method with RKD electrons is commendable, considering both ions and electrons as suprathermal (following RKD) could enhance the model's generalizability. This approach allows the model to address various plasma contexts by adjusting the spectral indices kappa_i and kappa_e, alpha_e, and alpha_i. I recommend revising the model to incorporate RKD for both species.

**Reply:** We are very grateful for your valuable suggestions. According to your suggestion, we have considered that both electrons and ions follow RKD. But it is very tedious to calculate at present. We are willing to explore it in our future work.

**Question 3:** An explicit analytical representation of RKD is crucial for clarity. Please define and explain terms such as the kappa index, alpha, and their implications on the distribution. For example, a high kappa index suggests proximity to thermal equilibrium. Include references for RKD formulation.

**Reply:** We appreciate for your constructive suggestion and quite agree that it is crucial to have an explicit analytical representation of RKD for clarity. We have added definitions and explanations of the implications of the spectral index $\kappa_e$ and cut-off parameter $\alpha$ on RKD and references in the manuscript. The revise is shown in page 3 line 27-Page 4 line 1-4.

**i.e.:** where $\kappa_e$, $\alpha$, and $U(a,c;x)$ are, respectively, the electron spectral index of

Kappa distribution, the exponential cutoff parameter and the Kummer function (Scherer et al., 2019; Oldham et al., 2009). For RKD, in order to retain the main features of the SKD, the exponential cutoff parameter $\alpha$ must be positive and much smaller than the unit (Liu, 2020; Zhou et al., 2022). The smaller electron spectral index $\kappa_e$ and cutoff parameter $\alpha$ mean that there are more energetic electrons (Liu et al., 2021). In fact, in the case of $\alpha \to 0$ and $\kappa_e > 1.5$, Eq. (5) is transformed into the SKD. When $\alpha \to 0$ and $\kappa_e \to \infty$, Eq. (5) reduces to the MD.

**Question 4:** To enhance the manuscript's credibility, outline the derivation steps or provide references for the RKD expression used. Adding an appendix for this purpose would be beneficial. For example, please explain how the authors obtained equation 5.

**Reply:** Your comments are constructive and valuable. With your insightful suggestion, we have added Appendices A and B on pages 14-16, where Appendix A is a detailed derivation of the electron density and Appendix B is a detailed derivation of the trapped ions distribution function.

**Question 5:** Also, provide a reference to Kummer function such that the reader who are interested could learn more about the function.

**Reply:** We sincerely appreciate your valuable comments. We have added two references for interested readers to learn more about the Kummer function on page 3 line 27.

**i.e.:** where $\kappa_e$, $\alpha$, and $U(a,c;x)$ are, respectively, the electron spectral index of Kappa distribution, the exponential cutoff parameter and the Kummer function (Scherer et al., 2019; Oldham et al., 2009).

**Question 6:** You mentioned that "Nevertheless, the distribution of electrons with the SKD in the potential field will be valid within the region where $\kappa_e$ is greater than 0 and alpha not equal to 0. In this paper, we extended the SKD to the range of $\kappa > 0$ to demonstrate the impact of the cut-off parameter α on the properties of ion BGK holes"- In the case of RKD to become SKD the kappa index should be greater than 3/2 and alpha should be almost 0. Given this is the case, your statement lacks precision.

**Reply:** Thank you for your comments. You are right. We have modified the statement "Nevertheless, the distribution of electrons with the SKD in the potential field will be valid within the region where $\kappa_e$ is greater than 0 and alpha not equal to 0. In this

paper, we extended the SKD to the range of $\kappa > 0$ to demonstrate the impact of the cut-off parameter $\alpha$ on the properties of ion BGK holes" into "However, the distribution of electrons with the RKD will be effective within the entire region where $\kappa_e$ is greater than 0 and $\alpha \neq 0$. It means that RKD can be used to fit the particles with a low kappa distribution, where low kappa represents $0 < \kappa_e < 3/2$. In this paper, we will investigate the effect of the cut-off parameter $\alpha$ and whole region of electron spectral index $\kappa_e$ (i.e., $\kappa_e > 0$) on the nature of ion BGK holes.". (page 4 line 4-7)

**Question 7:** In the section 3, the authors talk about equations 19, and 20. I did not see any equations 19 and 20.

**Reply:** Thank you so much for your careful works and we sincerely apologize for our carelessness. We have corrected the 'In this section, the plasma parameters related to ion BGK holes are examined by analyzing Eqs. (19) and (20)' into 'In this section, the plasma parameters related to ion BGK holes are examined by analyzing Eqs. (17) and (18)'. (Page 6 line 1)

**Question 8:** Another important factor that the authors should take into account is that when you make the figures all the axis should be fixed, then only the readers can understand the effects of various parameters. Please fix the axes of figures 1,2 and 3.

**Reply:** We appreciate for your constructive suggestion. With your valuable suggestion, we have fixed the axes of Figures 1, 2, and 3 and hope that this will help the reader to understand the effects of various parameters.
**i.e.:**
Fig. 1

[Figure]

[Figure]

Fig. 2

[Figure]

Fig. 3

**Question 9:** I completely agree with the physical explanation that you provided for the figures 1,2,3. But it should clearly mention that this clearly depends on the shape of the potential. So it should be written in such a way that for a fixed potential, ….

**Reply:** Thank you for your constructive comments. You are right, it should be pointed

out that this clearly depends on the shape of the potential. We have added it on page 6 line 10-12, 15-19.

**i.e.:** It can be observed from Figs. 1, 2 and 3 that the depth of the IHs becomes greater in response to an increased $\kappa_e$ for a fixed potential and $\alpha$.

When the potential and $\alpha$ are fixed, the lower the electron spectral index $\kappa_e$, the higher the concentration of superthermal electrons, then the higher the amount of energy it carries. As a result, more energy is transferred to the ions, then the ions have enough energy to escape the potential well and not be trapped, resulting in the inhibition of the formation of ion holes, so the formed holes are shallower. In addition, in Figs. 1, 2 and 3, it can also be clearly seen that when $\alpha$ increases, the IHs becomes deeper for a fixed potential and $\kappa_e$.

**Question 10:** Please check the caption of figure 4. It is always better if you keep the legends similar. I don't see any pink dashed or black line in fig 4 (b). So I encourage the authors to verify and modify the caption accordingly.

**Reply:** Thank you for your comments. According to your suggestion, we have redrawn figure 4 and modified the caption of figure 4.
**i.e.:**
Fig. 4

[Figure]

[Figure]

[Figure]

Caption: The trapped ion distribution function $f_{tr}(w)$ vs energy $w$ for different $\kappa_e$ and $\alpha$, while keeping $\delta = 0.2$ and $\psi = 2$ as fixed. (a) $\alpha = 0$, (b) $\alpha = 0.01$, (c) $\alpha = 0.2$, where the black, purple, blue, magenta, brown and orange line correspond to $\kappa_e = 3$, 100, 0.3, 1, 3 and 100, respectively.

**Question 11:** Also please explain the limits of alpha that appears in RKD. What happens if the value of alpha is large.

**Reply:** The cut-off parameter $\alpha$ must be positive and much less than 1 so that the RKD retains the main features of the SKD and is consistent with the observed superthermal tails. When $\alpha \to 0$ and $\kappa > 1.5$, the RKD is transformed into the SKD. When $\alpha \to 0$ and $\kappa \to \infty$, the RKD reduces to the MD.

When the cut-off parameter $\alpha$ is larger, there are fewer energetic electrons, and fewer energetic electrons will not be able to allow more ions to escape the potential well and become passing ions. As a result, more ions will be trapped, resulting in deeper ion holes.

What's more, there are other revisions in the manuscript.

**i.e.:**

page 9 line 2-5: As shown in Fig. 4, when $w$ approaches 0, the curves converge to the point of $f_{tr}(w) = 0.58$. In Figs. 4(a) and 4(b), when $\alpha = 0$, the distribution functions for $\kappa_e = 3$ and $\kappa_e = 100$ almost coincide with the distribution function when

$\alpha = 0.01$. In Fig. 4, the difference between $f_{tr}$ corresponding to $w = -0.1$ and the maximum value of $f_{tr}$ represents the depth of the IHs formed, from which it can be found that a larger $\kappa_e$ leads to deeper IHs when $\alpha$ is fixed.

page 10 line 2-3: It can also be seen that for the fixed $\kappa_e$, the larger $\alpha$ is, the deeper the IHs are formed.

page 12 line 1-3: It can be seen from Figs. 5(a) and 5(b) that for $\alpha = 0$, the curves of $\kappa_e = 3$ and $\kappa_e = 100$ are almost identical with $\alpha = 0.01$. In the case of a smaller $\kappa_e$, it can be observed that the plausible range increases for the fixed $\alpha$.

page 12 line 5-6: One can also find that when $\alpha$ is larger, the observed plausible region is smaller for the fixed $\kappa_e$.

We have redrawn figure 5 and modified the caption of figure 5.

**i.e.:**

[Figure]

[Figure]

[Figure]

**Caption:** The variation of width $\delta$ vs amplitude $\psi$ for different $\kappa_e$ and $\alpha$. (a) $\alpha = 0$, (b) $\alpha = 0.01$, (c) $\alpha = 0.2$, where the black, purple, blue, magenta, brown and orange line correspond to $\kappa_e = 3$, 100, 0.3, 1, 3 and 100, respectively.

The detailed description of the revisions made in the paper is included in the response. For more clearly reading, the highlighted manuscript where all the revises are marked with yellow color is also delivered.

All detail revises, please see the revised manuscript (the PDF file) where all the revises are marked with yellow color. We appreciate for Editors/Reviewers' warm work earnestly, and hope that the revisions will meet with approval.
Once again, we highly appreciate for your time and consideration.

Sincerely,

Qiuping Lu, Caiping Wu, Hui Chen, Xiaochang Chen and Sanqiu Liu
Corresponding author: Hui Chen
E-Mail: hchen61@ncu.edu.cn

---

## Author Comment (AC2)

**Letter to the editor**

Dear editor,

Thank you very much for the efficient work on the manuscript entitled "The dynamic of ion Bernstein–Greene–Kruskal holes in plasmas with regularized $\kappa$-distributed electrons" (npg-2023-25). We have revised the manuscript according to the Referees' comments. The detailed description of the revisions made in the paper is included in the response. For more clearly reading, the highlighted manuscript where all the revises are marked with yellow is also delivered. We appreciate for your warm work earnestly, and hope our revisions are acceptable. If you have any queries, please don't hesitate to contact us.

Thank you again.
Yours sincerely,
Qiuping Lu, Caiping Wu, Hui Chen, Xiaochang Chen and Sanqiu Liu
* * *
**Reply to Referees**

Dear Referee,

Deeply thanks to you for your constructive and very helpful comments. Our manuscript has been revised according to your comments, and the point-by-point replies for your comments are as follows. Detailed revises are marked with yellow in the revised version of manuscript. We appreciate for your warm work earnestly, and hope that the correction will meet with approval.

Thank you again for your comments and suggestions

Sincerely,
Qiuping Lu, Caiping Wu, Hui Chen, Xiaochang Chen and Sanqiu Liu.

**Point-by-point replies to the referee 2's comments as follows,**

**Question 1:** Although the paper title and abstract are about "The dynamic of ion Bernstein-Greene-Kruskal holes", this paper does not consider any dynamical properties and entirely focuses on construction of the stationary solutions of ion holes. There are multiple such solutions already published for different plasma distributions, and all these solutions may be important only in context of comparison with observations or investigation of hole dynamics and stability analysis. It's quite hard, if

possible, to justify construction of 1D electrostatic equilibrium without any analysis of applicability of this equilibrium to some realistic (observed in space or laboratory) structures. Therefore, the motivation for this study, and importance of obtained results are unclear.

**Reply:** We are very grateful for your valuable suggestions.
Firstly, if permitted by editorial office, we would like to change the title to make it more relevant to the content, according to your suggestion.
Secondly, the manuscript focuses on the space structure of ion holes, trapped ion distribution function-energy ($f_{tr} - w$), and width-amplitude ($\delta - \psi$) relation. The effects of the cut-off parameter $\alpha$ and spectral index $\kappa_e$ on the depth and energy of ion holes, and the allowed combination of width and amplitude to support physically plausible ion holes equilibrium are studied. The cut-off parameter $\alpha$ and spectral index $\kappa_e$ affect the number of superthermal electrons, which in turn affects the energy conversion between trapped ions and superthermal electrons and has an impact on the depth of ion holes and allowed combination of width and amplitude to support physically plausible ion holes equilibrium.

Thirdly, as for your comment of "There are multiple such solutions already published for different plasma distributions, and all these solutions may be important only in context of comparison with observations or investigation of hole dynamics and stability analysis. It's quite hard, if possible, to justify construction of 1D electrostatic equilibrium without any analysis of applicability of this equilibrium to some realistic (observed in space or laboratory) structures. Therefore, the motivation for this study, and importance of obtained results are unclear.", we fully agree with you that all these solutions may be important only when compared to observations or when studying hole dynamics and stability analysis and it's quite hard to justify construction of 1D electrostatic equilibrium without any analysis of applicability of this equilibrium to some realistic (observed in space or laboratory) structures. We are sorry that, at the moment, we have not found a concrete example of an observation and specific instances in space plasma environments where this condition is observed, but I hope that this study will provide a theoretical reference for the results of observing such a situation in space plasma, astrophysical plasma and laboratory.

**Question 2:** Figures 2, 3 show absolutely identical structures that are different only by color bars…

**Reply:** Thank you for your comments. We need to apologize for not expressing it clearly. Figure 2 denotes the phase space structure of the trapped ion distribution function $f_{tr}(x,v)$ in $x-v$ space for different $\kappa_e$ at $\alpha = 0.01$, $\delta = 0.2$ and

$\psi = 2$. While figure 3 represents the phase space structure of the trapped ion distribution function $f_{tr}(x,v)$ in $x-v$ space for different $\kappa_e$ at $\alpha = 0.2$, $\delta = 0.2$ and $\psi = 2$. The color bars in both figures 2 and 3 represent the depth of ion holes. It can be seen that when $\alpha$ increases, the IHs becomes deeper for a fixed potential and $\kappa_e$. When $\kappa_e$ increases, the IHs becomes deeper for a fixed potential and $\alpha$.

**Question 3:** Section 2 ends by Eq. (18), whereas Section 3 starts with "…BGK holes are examined by analyzing Eqs. (19) and (20)." I did not find these equations in the text…

**Reply:** Thank you so much for your careful works and we sincerely apologize for our carelessness. We have modified it in revised manuscript.
**i.e.:** In this section, the plasma parameters related to ion BGK holes are examined by analyzing Eqs. (17) and (18). (Page 6 line 1)

The detailed description of the revisions made in the paper is included in the response. For more clearly reading, the highlighted manuscript where all the revises are marked with yellow color is also delivered.

All detail revises, please see the revised manuscript (the PDF file) where all the revises are marked with yellow color. We appreciate for Editors/Reviewers' warm work earnestly, and hope that the revisions will meet with approval.
Once again, we highly appreciate for your time and consideration.

Sincerely,

Qiuping Lu, Caiping Wu, Hui Chen, Xiaochang Chen and Sanqiu Liu
Corresponding author: Hui Chen
E-Mail: hchen61@ncu.edu.cn